# Augmented Language Models: a Survey

**Grégoire Mialon**[*]                                                          *gmialon@meta.com*
**Roberto Dessì**[*†]                                                           *rdessi@meta.com*
**Maria Lomeli**[*]                                                             *marialomeli@meta.com*
**Christoforos Nalmpantis**[*]                                                  *christoforos@meta.com*
**Ram Pasunuru**[*]                                                            *rpasunuru@meta.com*
**Roberta Raileanu**[*]                                                         *raileanu@meta.com*
**Baptiste Rozière**[*]                                                         *broz@meta.com*
**Timo Schick**[*]                                                             *schick@meta.com*
**Jane Dwivedi-Yu**[*]                                                          *janeyu@meta.com*
**Asli Celikyilmaz**[*]                                                         *aslic@meta.com*
**Edouard Grave**[*]                                                            *egrave@meta.com*
**Yann LeCun**[*]                                                              *yann@meta.com*
**Thomas Scialom**[*]                                                          *tscialom@meta.com*

[*]*Meta AI*      [†]*Universitat Pompeu Fabra*

**Reviewed on OpenReview:** **https://openreview.net/forum?id=jh7wH2AzKK**

## Abstract

This survey reviews works in which language models (LMs) are augmented with reasoning skills and the ability to use tools. The former is defined as decomposing a potentially complex task into simpler subtasks while the latter consists in calling external modules such as a code interpreter. LMs can leverage these augmentations separately or in combination via heuristics, or learn to do so from demonstrations. While adhering to a standard missing tokens prediction objective, such augmented LMs can use various, possibly non-parametric external modules to expand their context processing ability, thus departing from the pure language modeling paradigm. We therefore refer to them as Augmented Language Models (ALMs). The missing token objective allows ALMs to learn to reason, use tools, and even act, while still performing standard natural language tasks and even outperforming most regular LMs on several benchmarks. In this work, after reviewing current advance in ALMs, we conclude that this new research direction has the potential to address common limitations of traditional LMs such as interpretability, consistency, and scalability issues.

## 1   Introduction: motivation for the survey and definitions

### 1.1   Motivation and Definitions

Large Language Models (LLMs) (Devlin et al., 2019; Brown et al., 2020; Chowdhery et al., 2022) have fueled dramatic progress in Natural Language Processing (NLP) and are already core in several products with millions of users, such as the coding assistant Copilot (Chen et al., 2021), Google search engine[1] or more recently ChatGPT[2]. Memorization (Tirumala et al., 2022) combined with compositionality (Zhou et al., 2022) capabilities made LLMs able to execute various tasks such as language understanding or conditional and

---

[1]See *e.g.* https://blog.google/products/search/search-language-understanding-bert/
[2]https://openai.com/blog/chatgpt/

unconditional text generation at an unprecedented level of performance, thus opening a realistic path towards higher-bandwidth human-computer interactions.

However, LLMs suffer from important limitations hindering a broader deployment. LLMs often provide non-factual but seemingly plausible predictions, often referred to as hallucinations (Welleck et al., 2020). This leads to many avoidable mistakes, for example in the context of arithmetics (Qian et al., 2022) or within a reasoning chain (Wei et al., 2022c). Moreover, many LLMs groundbreaking capabilities seem to emerge with size, measured by the number of trainable parameters: for example, Wei et al. (2022b) demonstrate that LLMs become able to perform some BIG-bench tasks[3] via few-shot prompting once a certain scale is attained. Although a recent line of work yielded smaller LMs that retain some capabilities from their largest counterpart (Hoffmann et al., 2022), the size and need for data of LLMs can be impractical for training but also maintenance: continual learning for large models remains an open research question (Scialom et al., 2022). Other limitations of LLMs are discussed by Goldberg (2023) in the context of *ChatGPT*, a chatbot built upon *GPT3*.

We argue these issues stem from a fundamental defect of LLMs: they are generally trained to perform statistical language modeling given (i) a single parametric model and (ii) a limited context, typically the $n$ previous or surrounding tokens. While $n$ has been growing in recent years thanks to software and hardware innovations, most models still use a relatively small context size compared to the potentially large context needed to always correctly perform language modeling. Hence, massive scale is required to store knowledge that is not present in the context but necessary to perform the task at hand.

As a consequence, a growing research trend emerged with the goal to solve these issues, slightly moving away from the pure statistical language modeling paradigm described above. For example, a line of work circumvents the limited context size of LLMs by increasing its relevance: this is done by adding information extracted from relevant external documents. Through equipping LMs with a module that retrieves such documents from a database given a context, it is possible to match certain capabilities of some of the largest LMs while having less parameters (Borgeaud et al., 2022; Izacard et al., 2022). Note that the resulting model is now non-parametric, since it can query external data sources. More generally, LMs can also improve their context via reasoning strategies (Wei et al. (2022c); Taylor et al. (2022); Yang et al. (2022c) *inter alia*) so that a more relevant context is produced in exchange for more computation before generating an answer. Another strategy is to allow LMs to leverage external tools (Press et al. (2022); Gao et al. (2022); Liu et al. (2022b) *inter alia*) to augment the current context with important missing information that was not contained in the LM's weights. Although most of these works aim to alleviate the downfalls of LMs mentioned above separately, it is straightforward to think that more systematically augmenting LMs with both reasoning and tools may lead to significantly more powerful agents. We will refer to these models as **Augmented Language Models (ALMs)**. As this trend is accelerating, keeping track and understanding the scope of the numerous results becomes arduous. This calls for a taxonomy of ALMs works and definitions of technical terms that are used with sometimes different intents.

**Definitions.** We now provide definitions for terms that will be used throughout the survey.

- **Reasoning.** In the context of ALMs, reasoning is decomposing a potentially complex task into simpler subtasks the LM can solve more easily by itself or using tools. There exist various ways to decompose into subtasks, such as recursion or iteration. In that sense, reasoning is akin to planning as defined for example in LeCun (2022). In this survey, reasoning will very often refer to the various strategies to improve reasoning skills in LMs, such as step-by-step reasoning using few-shot examples. It is not yet fully understood whether the LM is really reasoning, or simply producing a larger context that increases the likelihood of correctly predicting the missing tokens. We refer to Huang and Chang (2022) for a discussion on this topic: although reasoning may currently be an abuse of language given the current state of the art, the term is already in use within the community. A more pragmatic definition of reasoning in the context of ALMs is giving more computation steps to the model before yielding the answer to a prompt.

---

[3] https://github.com/google/BIG-bench

- **Tool.** For ALMs, a tool is an external module that is typically called using a rule or a special token and whose output is included in the ALM's context. The tool can gather external information, or have an effect on the virtual or physical world (generally perceived by the ALM). An example of a tool fetching external information is a document retriever, while a tool having an external effect could be a robotic arm. A tool can be called at training or at inference time. More generally, learning to interact with a tool may consist in learning to call its API.

- **Act.** For ALMs, calling a tool that modifies a state in a virtual or physical object, and observing the result, typically by including it in the ALM's current context. For example, some works from the survey discuss searching the web, or robotic arm manipulation via LMs. With a slight abuse of term, we will sometimes denote the call of a tool by an ALM as an action, even if it does not have an external effect.

**Why jointly discussing reasoning and tools?** The combination of reasoning and tools within LMs should allow solving a broad range of complex tasks without heuristics, hence with better generalization capabilities. Typically, reasoning would foster the LM to decompose a given problem into potentially simpler subtasks while tools would help getting each step right, for example obtaining the result from a mathematical operation. Put it differently, reasoning is a way for LMs to combine different tools in order to solve complex tasks, and tools are a way to not fail a reasoning with valid decomposition. Both should benefit from the other. Moreover, reasoning and tools can be put under the same hood, as both augment the context of the LM so that it better predicts the missing tokens, albeit in a different way.

**Why jointly discussing tools and actions?** Tools that gather additional information and tools that have an effect on the virtual or physical world can be called in the same fashion by the LM. For example, there is seemingly no difference between a LM outputting python code for solving a mathematical operation, and a LM outputting python code to manipulate a robotic arm. A few works discussed in the survey are already using LMs that have effects on the virtual or physical world: under this view, we can say that the LM have the potential to act, and expect important advances in the direction of LMs as autonomous agents.

## 1.2 Our classification

We decompose the works included in the survey under three axes. Section 2 studies works which augment LM's reasoning capabilities as defined above. Section 3 focuses on works allowing LMs to interact with external tools and act. Finally, Section 4 explores whether reasoning and tools usage are implemented via heuristics or learned, *e.g.* via supervision or reinforcement. Other axes could naturally have been chosen for this survey and are discussed in Section 5. For conciseness, the survey focuses on works that combine reasoning or tools with LMs. However, the reader should keep in mind that many of these techniques were originally introduced in another context than LMs, and consult the introduction and related work section of the papers we mention if needed. Finally, although we focus on LLMs, not all works we consider employ large models, hence we stick to LMs for correctness in the remainder of the survey.

## 2 Reasoning

In general, reasoning is the ability to make inferences using evidence and logic. Reasoning can be divided into multiple types of skills such as commonsense reasoning (McCarthy et al., 1960; Levesque et al., 2012), mathematical reasoning (Cobbe et al., 2021), symbolic reasoning (Wei et al., 2022c), etc. Often, reasoning involves deductions from inference chains, called as multi-step reasoning. In the context of LMs, we will use the definition of reasoning provided in Section 1. Previous work has shown that LLMs can solve simple reasoning problems but fail at complex reasoning (Creswell et al., 2022): hence, this section focuses on various strategies to augment LM's reasoning skills. One of the challenges with complex reasoning problems for LMs is to correctly obtain the solution by composing the correct answers predicted by it to the sub-problems. For example, a LM may correctly predict the dates of birth and death of a celebrity, but may not correctly predict the age. Press et al. (2022) call this discrepancy the compositionality gap for LMs. For the rest of this section, we discuss the works related to three popular paradigms for eliciting reasoning in LMs. Note

that Huang and Chang (2022) propose a survey on reasoning in language models. Qiao et al. (2022) also propose a survey on reasoning albeit with a focus on prompting. Since our present work focuses on reasoning combined with tools, we refer the reader to Huang and Chang (2022); Qiao et al. (2022) for a more in-depth review of works on reasoning for LLMs.

## 2.1 Eliciting reasoning with prompting

In recent years, prompting LMs to solve various downstream tasks has become a dominant paradigm (Brown et al., 2020). In prompting, examples from a downstream task are transformed such that they are formulated as a language modeling problem. Prompting typically takes one of the two forms: zero-shot, where the model is directly prompted with a test example's input; and few-shot, where few examples of a task are prepended along with a test example's input. This few-shot prompting is also known as in-context learning or few-shot learning. As opposed to "naive" prompting that requires an input to be directly followed by the output/answer, elicitive prompts encourage LMs to solve tasks by following intermediate steps before predicting the output/answer. While Nye et al. (2021) provides the first example of few-shot prompting LLMs with reasoning examples and Cobbe et al. (2021) generalizes the use of reasoning examples to non-algorithmic tasks, Wei et al. (2022c) extensively studies how elicitive prompting enables LMs to be better reasoners in a few-shot setting. Later, Kojima et al. (2022) showed similar ability in a zero-shot setting. We discuss them in detail in the following paragraphs.

**Few-shot setting.** Wei et al. (2022c) popularized chain-of-thought (CoT), a few-shot prompting technique for LMs. The prompt consists of examples of a task, with inputs followed by intermediate reasoning steps leading to the final output, as depicted in Figure 1. Table 1 shows that CoT outperforms standard prompting methods. Wei et al. (2022b) observe that the success of the few-shot strategy emerges with scale, while Tay et al. (2022) add that without fine-tuning, successful use of CoT generally requires 100B+ parameters LMs such as *LaMDA* (Thoppilan et al., 2022), *PaLM* (Chowdhery et al., 2022) or *GPT3* (Brown et al., 2020; Ouyang et al., 2022), before proposing *UL2*, a 20B open source model that can perform CoT. Using few-shot CoT prompting, *Minerva* (Lewkowycz et al., 2022) achieves excellent performance on math benchmarks such as GSM8K (Cobbe et al., 2021). Wang et al. (2022c) further improve CoT with *Self-consistency*: diverse reasoning paths are sampled from a given language model using CoT, and the most consistent answer is selected as the final answer. Press et al. (2022) introduce *Self-ask*, a prompt in the spirit of CoT. Instead of providing the model with a continuous chain of thought as in Figure 1, *Self-ask* explicitly states the follow-up question before answering it and relies on a scaffold (e.g, *"Follow-up question:"* or *"So the final answer is:"*), so that the answers are more easily parseable. The authors demonstrate an improvement over CoT on their introduced datasets aiming at measuring the compositionality gap. They observe that this gap does not narrow when increasing the size of the model. Note that Press et al. (2022) focus on 2-hop questions, *i.e.*, questions for which the model only needs to compose two facts to obtain the answer. Interestingly, *Self-ask* can easily be augmented with a search engine (see Section 3). *ReAct* (Yao et al., 2022b) is another few-shot prompting approach eliciting reasoning that can query three tools throughout the reasoning steps: `search` and `lookup` in Wikipedia, and `finish` to return the answer. *ReAct* will be discussed in more detail in the next sections.

**Zero-shot setting.** Kojima et al. (2022) extend the idea of eliciting reasoning in LMs to zero-shot prompting. Whereas few-shot provides examples of the task at hand, zero-shot conditions the LM on a single prompt that is not an example. Here, Kojima et al. (2022) simply append *Let's think step by step* to the input question before querying the model (see Figure 2), and demonstrate that zero-shot-CoT for large LMs does well on reasoning tasks such as GSM8K although not as much as few-shot-CoT.

## 2.2 Recursive prompting

Several works attempt to improve LM's reasoning by explicitly decomposing problems into sub-problems in order to solve the problem in a divide and conquer manner. This recursive approach slightly differs from CoT since the latter does not explicitly formulate sub-problems, and can be especially useful for complex tasks, given that compositional generalization can be challenging for LMs (Lake and Baroni, 2018; Keysers et al.,

> **Question:** Roger has 5 tennis balls. He buys 2 more cans of tennis balls. Each can has 3 tennis balls. How many tennis balls does he have now?
> **Answer:** Roger started with 5 balls. 2 cans of 3 tennis balls each is 6 tennis balls. $5 + 6 = 11$. The answer is 11.
> **Question:** The cafeteria had 23 apples. If they used 20 to make lunch and bought 6 more, how many apples do they have?
> **Answer:**
> **\<LM\>**

Figure 1: An example of few-shot Chain-of-Thought prompt. **\<LM\>** denotes call to the LM with the above prompt.

> **Question:** The cafeteria had 23 apples. If they used 20 to make lunch and bought 6 more, how many apples do they have?
> **Answer:** Let's think step by step
> **\<LM\>**

Figure 2: An example of zero-shot Chain-of-Thought prompt. **\<LM\>** denotes call to the LM with the above prompt.

2019; Li et al., 2022a). Methods that employ problem decomposition can either then solve the sub-problems independently, where these answers are aggregated to generate the final answer (Perez et al., 2020; Min et al., 2019), or solve the sub-problems sequentially, where the solution to the next sub-problem depends on the answer to the previous ones (Yang et al., 2022a; Zhou et al., 2022; Drozdov et al., 2022; Dua et al., 2022; Khot et al., 2022; Wang et al., 2022a; Wu et al., 2022b; Mishra and Nouri, 2022). For instance, in the context of math problems, *Least-to-most* prompting (Zhou et al., 2022) allows a language model to solve harder problems than the demonstration examples by decomposing a complex problem into a list of sub-problems. It first employs few-shot prompting to decompose the complex problem into sub-problems, before sequentially solving the extracted sub-problems, using the solution to the previous sub-problems to answer the next one. Patel et al. (2022) show that modifying existing benchmarks by letting human decompose questions into subquestions that are relatively easier for models to solve leads to significant improvement for *GPT3* and *RoBERTa-SQuAD* equipped with a symbolic calculator.

While many earlier works include learning to decompose through distant supervision (Perez et al., 2020; Talmor and Berant, 2018; Min et al., 2019), like Zhou et al. (2022), many recent works employ in-context learning to do so (Yang et al., 2022a; Khot et al., 2022; Dua et al., 2022; Kazemi et al., 2022). Among these, there are further differences. For instance, Drozdov et al. (2022) is a follow-up work to Zhou et al. (2022), but differs by using a series of prompts to perform recursive syntactic parses of the input rather than a linear decomposition, and also differs by choosing the exemplars automatically through various heuristics. Dua et al. (2022) is concurrent work with Zhou et al. (2022) but differs by interweaving the question decomposition and answering stages, i.e., the next sub-question prediction has access to the previous questions and answers as opposed to generating all sub-questions independently of any previous answers. Yang et al. (2022a), on the other hand, decomposes using rule-based principles and slot-filling prompting to translate questions into a series of SQL operations. Khot et al. (2022) also employs prompts to decompose into specific operations, but then allows each sub-problem to be solved using a library of specialized handlers, where each is devoted to a particular sub-task (e.g., retrieval). Finally, Kazemi et al. (2022) decompose a given problem in a backward fashion: starting from the goal and a set of rules, the system decompose the goal into sub-goals and recursively check whether the sub-goals can be proved. Here again, the modules are implemented by few-shot prompting a pre-trained LM.

| Model | Accuracy (%) |
|---|---|
| OpenAI (`text-davinci-002`)[1] | 15.6 |
| OpenAI (`text-davinci-002`) + CoT[1] | 46.9 |
| OpenAI (`text-davinci-002`) + CoT + Calculator[1] | 46.9 |
| OpenAI (`code-davinci-002`)[1] | 19.7 |
| OpenAI (`code-davinci-002`) + CoT[1] | 63.1 |
| OpenAI (`code-davinci-002`) + CoT + Calculator[1] | 65.4 |
| GPT-3 175B + FT + CoT + Calculator[2] | 34.0 |
| GPT-3 175B + FT + CoT + Calculator + Verifier[2] | 55.0 |
| PaLM 540B[3] | 17.0 |
| PaLM 540B+CoT[3] | 54.0 |
| PaLM 540B+CoT+Calculator[3] | 58.0 |
| PAL[4] | 72.0 |

Table 1: Evaluation of different reasoning methods on GSM8K, a popular reasoning benchmark. FT denotes fine-tuning and CoT denotes chain-of-thought. The reported accuracies are based on [1]: (Wei et al., 2022c); [2]: (Cobbe et al., 2021); [3]: (Chowdhery et al., 2022); and [4]: (Gao et al., 2022).

## 2.3 Explicitly teaching language models to reason

Despite their spectacular results, prompting approaches have some drawbacks in addition to requiring model scale. Namely, they require to discover prompts that elicit e.g. step-by-step reasoning, manually providing examples when it comes to few-shot for a new task. Moreover, prompting is computationally expensive in the case of long prompts, and it is harder to benefit from a relatively large number of examples due to limited context size of the model. Recent works suggest to circumvent these issues by training LMs to use, as humans, a working memory when more than one step are required to solve a task correctly. Nye et al. (2021) introduce the notion of scratchpad, allowing a LM to better perform on multi-step computation tasks such as addition or code execution. More precisely, at training time, the LM sees input tasks such as addition along with associated intermediate steps: the ensemble is called a scratchpad. At test time, the model is required to predict the steps and the answer from the input task. Scratchpads differ from the above prompting strategies in that they are fine-tuned on example tasks with associated computation steps. Note however that Nye et al. (2021) also perform experiments in the few-shot regime. Taylor et al. (2022) use a similar approach in the context of large LM pre-training: *Galactica* was trained on a corpus of scientific data including some documents where step-by-step reasoning is wrapped with a special token `<work>` and `</work>` to mimic an internal working memory. At inference time, the model can be asked explicitly to activate this reasoning mode via the `<work>` token. Taylor et al. (2022) argue that one more problem arise when training on reasoning examples: many intermediate reasoning steps may be missing in the training data curated from the internet, as humans do not explicitly write all their reasoning steps. To circumvent the issue of missing steps, the authors created datasets with detailed reasoning process. An example of prompt seen during *Galactica*'s pre-training is presented in Figure 4.

Other recent works improve the reasoning abilities of pre-trained LMs via fine-tuning. Zelikman et al. (2022) propose a bootstrap approach to generate reasoning steps (also called rationales) for a large set of unlabeled data and use that data to fine-tune the model. Lengerich et al. (2022) propose a self-supervised method which extracts reasoning capabilities of a teacher model by asking the question "why" given the student's initial responses to various NLP problems. Next, the student model is fine-tuned by using contrastive distillation via sampling evidence from memory. Yu et al. (2022) show that standard LM fine-tuning on reasoning tasks lead to better reasoning skills such as textual entailment, abductive reasoning, and analogical reasoning, compared to pre-trained models. Further, several instruction fine-tuning approaches (Ouyang et al., 2022; Chung et al., 2022; Iyer et al., 2022; Ho et al., 2022) use chain-of-thought style prompts to achieve remarkable improvements on popular benchmarks such as BBH (Srivastava et al., 2022) and MMLU (Hendrycks et al., 2021). Interestingly, all these works also show that small scale instruction-finetuned models can perform better than un-finetuned large scale models, especially in the tasks where instruction following is important.

---

**Prompt 0**

**Question:** It takes Amy 4 minutes to climb to the top of a slide. It takes her 1 minute to slide down. The water slide closes in 15 minutes. How many times can she slide before it closes?
**<LM>**
**Answer:** To solve " How many times can she slide before it closes? ", we need to first solve: " How long does each trip take? "
**</LM>**

**Prompt 1**

It takes Amy 4 minutes to climb to the top of a slide. It takes her 1 minute to slide down. The water slide closes in 15 minutes.
**Subquestion 1**: How long does each trip take?
**<LM>**
**Answer 1**: It takes Amy 4 minutes to climb and 1 minute to slide down. $4 + 1 = 5$. So each trip takes 5 minutes.
**</LM>**

**Prompt 2**

It takes Amy 4 minutes to climb to the top of a slide. It takes her 1 minute to slide down. The slide closes in 15 minutes.
**Subquestion 1**: How long does each trip take?
**Answer 1**: It takes Amy 4 minutes to climb and 1 minute to slide down. $4 + 1 = 5$. So each trip takes 5 minutes.
**Subquestion 2**: How many times can she slide before it closes?
**<LM>**
**Answer 2**: The water slide closes in 15 minutes. Each trip takes 5 minutes. So Amy can slide $15 \div 5 = 3$ times before it closes.
**</LM>**

Figure 3: Recursive prompting example. **<LM>** denotes the start of the LM's output to the prompt, while **</LM>** denotes the end. The problem is first decomposed into subproblems in **Prompt 0**. Then, **Answer 2** to **Subquestion 2** and **Answer 1** to **Subquestion 1** are sequentially fed to **Prompt 2** and **Prompt 1**. The few-shot examples for each stage's prompt are omitted. Inspired from Figure 1 in Zhou et al. (2022).

## 2.4 Comparison and limitations of abstract reasoning

Overall, reasoning can be seen as decomposing a problem into a sequence of sub-problems either iteratively or recursively.[4] Exploring as many reasoning paths as possible is hard and there is no guarantee that the intermediate steps are valid. A way to produce faithful reasoning traces is to generate pairs of questions and their corresponding answers for each reasoning step (Creswell and Shanahan, 2022), but there is still no guarantee of the correctness of these intermediate steps. Overall, a reasoning LM seeks to improve its context by itself so that it has more chance to output the correct answer. To what extent LMs actually use the stated reasoning steps to support the final prediction remains poorly understood (Yu et al., 2022).

---

[4]Here, reasoning is described as a sequential operation. However, other reasoning structures such as trees could be considered. For example, Lample et al. (2022) leverage trees to model the different strategies leading to a proof for a given theorem. A strategy is a set of intermediate results that must be either true or themselves proved, hence decomposed into another new subset of intermediate results.

**Question:** A needle 35 mm long rests on a water surface at $20 \circ$ C. What force over and above the needle's weight is required to lift the needle from contact with the water surface? $\sigma = 0.0728m$.
`<work>`

$$\sigma = 0.0728N/m$$
$$\sigma = F/L$$
$$0.0728 = F/(2 \times 0.035)$$
$$F = 0.0728(2 \times 0.035)$$

`calculate.py`
```
" '
f = 0.0728*(2*0.035)
with open("output.txt", "w") as file:
file.write(str(round(f, 5)))
" ,
```

«run: `calculate.py`»

«read: `output.txt`»

0.0051

`</work>`

**Answer:** $F = 0.0051N$

Figure 4: Working memory example from Taylor et al. (2022). This prompt and its output are seen during LM pre-training.

In many cases, some reasoning steps may suffer from avoidable mistakes that compromise the correctness of the output. For example, mistakes on nontrivial mathematical operations in a reasoning step may lead to the wrong final output. The same goes with known facts such as the identity of a president at a given year. Some of the works studied above (Yao et al., 2022b; Press et al., 2022) already leverage simple external tools such as a `search engine` or a `calculator` to validate intermediate steps. More generally, the next section of the survey focuses on the various tools that can be queried by LMs to increase the chance of outputting a correct answer.

## 3 Using Tools and Act

A recent line of LM research allows the model to access knowledge that is not necessarily stored in its weights, such as a given piece of factual knowledge. More precisely, tasks such as exact computation or information retrieval for example can be offloaded to external modules such as a `python interpreter` or a `search engine` that are queried by the model which, in that respect, use tools. Additionally, we can say the LM performs an action when the tool has an effect on the external world. The possibility to easily include tools and actions in the form of special tokens is a convenient feature of language modeling coupled with transformers.

### 3.1 Calling another model

In many cases, the tool can simply be another neural network or the LM itself.

**Iteration 0**

**Text:** Brittney Reese (born September 9, 1986 in Gulfport, Mississippi) is an American long jumper.
**<LM>**
**Plan:** Remove incorrect information
**Edit:** Brittney Reese (born September 9, 1986 ~~in Gulfport, Mississippi~~) is an American long jumper.
**</LM>**

**Iteration 1**

**Text:** Brittney Reese (born September 9, 1986) is an American long jumper.
**<LM>**
**Plan:** Add information about her career
**Edit:** Brittney Reese (born September 9, 1986) is an American long jumper , who competed at the 2008 Summer Olympics, and is a 4-time World Champion .
**</LM>**

**Iteration 2**

**Text:** Brittney Reese (born September 9, 1986) is an American long jumper, who competed at the 2008 Summer Olympics, and is a 4-time World Champion.
**<LM>**
**Plan:** Add her birthplace
**Edit:** Brittney Reese (born September 9, 1986 in Inglewood, California ) is an American long jumper, who competed at the 2008 Summer Olympics, and is a 4-time World Champion.
**</LM>**

Figure 5: Iterative prompting example using PEER (Schick et al., 2022), a LM trained to produce a plan of action and edit to the input text at each step. This process can be repeated until the generated text requires no further updates. **<LM>** denotes the start of the LM's output to the prompt, while **</LM>** denotes the end.

**Iterative LM calling.** As an alternative to improving the LM's context for better outputs after a single inference pass, an alternative and intuitive way to get better results from LMs consists of repeatedly calling the model to iteratively refine its output. *Re3* (Yang et al., 2022c) exploits this idea to automatically generate stories of over two thousand words. More precisely, *Re3* first generates a plan, setting, and characters by prompting *GPT3* (Brown et al., 2020) with a premise. Then, *Re3* iteratively injects information from both the plan and current story state into a new *GPT3* prompt to generate new story passages. This work is improved upon in Yang et al. (2022b) with the use of a learned detailed outliner that iteratively expands the brief initial outline to any desired level of granularity. Other approaches that teach models to iteratively improve texts in an unsupervised fashion range from applications such as blank filling (Shen et al., 2020; Donahue et al., 2020) to denoising a sequence of Gaussian vectors into word vectors (Li et al., 2022c). *PEER* (Schick et al., 2022), for example, is a model initialized from *LM-Adapted T5* (Raffel et al., 2020) and trained on Wikipedia edits, learning both how to carry out edits and how to plan for the next steps. Consequently, *PEER* is able to develop articles by repeatedly planning and editing as in Figure 5. The iterative approach has the additional benefit of allowing a complex task like story and article generation to be decomposed into smaller subtasks. Importantly and apart from *PEER*, the works mentioned above employ heuristics to call the LM. A future research direction may consist in allowing the LM to call itself repeatedly until the output satisfies a certain criterion. Rather than just calling a single model repeatedly, Wu et al.

(2022a) propose an interactive interface for a pipeline allowing chaining of multiple LMs together, where the output of one step is passed as input to the next. Such contributions allow non-AI-experts to refine solutions to complex tasks that cannot be appropriately handled by a single LM.

**Leveraging other modalities.** Prompts under the form of text may not contain enough context to correctly perform a given task. For example, a question does not call for the same answer if it is asked with a serious or ironic tone. Including various modalities into the context would probably be useful for LMs such as chatbots. As recently demonstrated by Hao et al. (2022) and Alayrac et al. (2022), LMs can also be used as a general-purpose interface with models pre-trained on different modalities. For example, Hao et al. (2022) take a number of pre-trained encoders that can process diverse modalities such as vision and language, and connect them to a LM that serves as a universal task layer. The interface and modular encoders are jointly pre-trained via a semi-causal language modeling objective. This approach combines the benefits of causal and non-causal language modeling, enabling both in-context learning and open-ended generation, as well as easy fine-tuning of the encoders. Similarly, Alayrac et al. (2022) introduce *Flamingo*, a family of Visual Language Models (VLMs) that can handle any interleaved sequences of visual and textual data. *Flamingo* models are trained on large-scale multimodal web corpora containing interleaved text and images, which enables them to display in-context few-shot learning capabilities of multimodal tasks. With only a handful of annotated examples, *Flamingo* can easily adapt to both generation tasks such as visual question-answering and captioning, as well as classification tasks such as multiple-choice visual question-answering. Zeng et al. (2022) introduce Socratic Models, a modular framework in which various models pre-trained on different modalities can be composed zero-shot. This allows models to exchange information with each other and acquire new multimodal capabilities without additional finetuning. Socratic Models enable new applications such as robot perception and planning, free-form question-answering about egocentric videos, or multimodal assistive dialogue by interfacing with external APIs and databases such as search engines. Interestingly, other modalities such as images can be incorporated to improve reasoning capabilities of moderate size LMs (1B) (Zhang et al., 2023), and enabling multimodal chain of thought reasoning (Lu et al., 2022a).

## 3.2 Information retrieval

LMs can be augmented with memory units, for example via a neural cache of recent inputs (Grave et al., 2017; Merity et al., 2017), to improve their reasoning abilities. Alternatively, knowledge in the form of natural language can be offloaded completely from the LM by retrieving from an external knowledge source. Memory augmentation strategies help the language model to avoid producing non-factual and out-of-date information as well as reducing the number of parameters required to achieve comparable performance to large LMs.

### 3.2.1 Retrieval-augmented language models

**Dense and sparse retrievers.** There exist two types of retrievers that can be used to augment a LM: dense and sparse. Sparse retrievers work with sparse bag-of-words representations of the documents and the queries (Robertson and Zaragoza, 2009). In contrast, dense neural retrievers use a dense query and dense document vectors obtained from a neural network (Asai et al., 2021). Both types of retrievers assess the relevance of a document to an information-seeking query. This can be done by (i) checking for precise term overlap or (ii) computing the semantic similarity across related concepts. Sparse retrievers excel at the first sub-problem, while dense retrievers can be better at the second (Luan et al., 2021).

**Conditioning LMs on retrieved documents.** Various works augment LMs with a `dense retriever` by adding the retrieved documents to the current context (Chen et al., 2017; Clark and Gardner, 2017; Lee et al., 2019; Guu et al., 2020; Khandelwal et al., 2020; Lewis et al., 2020; Izacard and Grave, 2020; Zhong et al., 2022; Borgeaud et al., 2022; Izacard et al., 2022; Shi et al., 2023). Even though the idea of retrieving documents to perform question answering is not new, retrieval-augmented LMs have recently demonstrated strong performance in other knowledge intensive tasks besides Q&A. These proposals close the performance gap compared to larger LMs that use significantly more parameters. *REALM* (Guu et al., 2020) was the first method to jointly train end-to-end a retrieval system with an encoder LM. *RAG* (Lewis et al., 2020) jointly fine-tunes the retriever with a sequence-to-sequence model. Izacard and Grave (2020) introduced a

modification of the seq2seq architecture to efficiently process many retrieved documents. Borgeaud et al. (2022) focuses on an auto-regressive LM, called *RETRO*, and use a large-scale corpus indexed by frozen *BERT* embeddings for the retriever module. Crucially, the authors show that at scale (retrieval corpus and language model), this approach does not require to train and update the retriever. In particular, *RETRO* obtains comparable performance to *GPT3* on different downstream tasks. Although *RETRO* was trained with retrieval from scratch, their approach allows the integration of retrieval into existing pre-trained LMs. *Atlas* (Izacard et al., 2022) jointly trains a retriever with a sequence-to-sequence model to obtain a LM with strong few-shot learning capabilities in spite of being orders of magnitude smaller than many other large LMs. Table 2 compares the main characteristics of the models discussed, notably how the retrieval results are integrated into the LM's context. In all the aforementioned cases, the query corresponds to the prompt but this has been relaxed, see the chain-of-thought subsection below.

| Model | # Retrieval tokens | Granularity | Retriever training | Retrieval integration |
|---|---|---|---|---|
| *REALM* (Guu et al., 2020) | $O(10^9)$ | Prompt | End-to-End | Append to prompt |
| *RAG* (Lewis et al., 2020) | $O(10^9)$ | Prompt | Fine-tuning | Cross-attention |
| *RETRO* (Borgeaud et al., 2022) | $O(10^{12})$ | Chunk | Frozen | Chunked cross-attn. |
| *Atlas* (Izacard et al., 2022) | $O(10^9)$ | Prompt | Fine-tuning | Cross-attention |

Table 2: Comparison between database retrieval augmented languages models. Inspired by Table 3 from Borgeaud et al. (2022).

**Efficient large scale retrieval.** Since retrieval-augmented language models store knowledge in an external data store, it is crucial that the information retrieval step is efficient, especially when dealing with a large number of document/passage/sentence embeddings and/or a large number (billions) of query vectors. The literature offers a variety of optimisations to boost performance of retrievers as well as reducing the memory footprint of the data store.

First, an index structure can be built for the document/passage/sentence embeddings to perform efficient similarity search, for instance, using the Facebook AI Similarity Search library (faiss) (Johnson et al., 2021). Leveraging indexing structures enables efficient search operations by partitioning the vector space and enabling fast pruning of irrelevant vectors.

Second, if there exist memory constraints due to a large number of document/passage/sentence embeddings, a multi-node, multi-gpu distributed framework can be used to store the index corresponding to only part of the document/passage/sentence embeddings per process. The query embeddings can also be split in batches and obtain the top-k results in parallel using gpus for further speed ups for approximate or exact search. When exact search becomes intractable due to scale, approximate nearest neighbor algorithms can be used. These algorithms trade off accuracy for speed, allowing significantly faster retrieval while maintaining reasonably accurate results, see Johnson et al. (2021) for further details.

Finally, in order to further reduce the memory footprint of each of the index shards corresponding to a subset of the document/passage/sentence embeddings, different compression techniques can be used. For instance, the Atlas retrieval-augmented language model (Izacard et al., 2022) uses product quantisation (Jégou et al., 2011) to reduce the memory footprint of the retriever without sacrificing accuracy in downstream tasks in terms of exact match and recall@50 metrics of Q&A.

Currently, there also exist a variety of vector database frameworks that have some of these optimisations implemented out of the box and can be leveraged to use retrieval as a service without requiring the user to implement all optimisations from scratch.

**Chain-of-thought prompting and retrievers.** Recent works (He et al., 2022; Trivedi et al., 2022) propose to combine a `retriever` with reasoning via chain-of-thoughts (CoT) prompting to augment a LM. He et al. (2022) use the CoT prompt to generate reasoning paths consisting of an explanation and prediction pair. Then, knowledge is retrieved to support the explanations and the prediction that is mostly supported by the evidence is selected. This approach does not require any additional training or fine-tuning. Trivedi et al. (2022) propose an information retrieval chain-of-thought approach (IRCoT) which consists of interleaving

retrieval with CoT for multi-step QA. The idea is to use retrieval to guide the CoT reasoning steps and conversely, using CoT reasoning to guide the retrieval step.

In all these works, a `retriever` is systematically called for every query in order to get the corresponding documents to augment the LM. These approaches also assume that the intent is contained in the query. The query could be augmented with the user's intent by providing a natural language description of the search task (instruction) in order to disambiguate the intent, as proposed by Asai et al. (2022). Also, the LM could query the retriever only occasionally—when a prompt suggests it to do so—which is discussed in the next subsection.

### 3.2.2   Querying search engines

In the previous paragraph, the information retrieval query corresponds to the LM's context. However, the LM can also have the ability to generate a query based on the prompt, thus enlarging its action space and becoming more active.

*LaMDA* is one example of an agent-like LM designed for dialogue applications. The authors pre-train the model on dialog data as well as other public web documents. In addition to this, to ensure that the model is factually grounded as well as enhancing its conversational abilities, it is augmented with `retrieval`, a `calculator`, and a `translator` (Thoppilan et al., 2022). Furthermore, to improve the model's safety, *LaMDA* is fine-tuned with annotated data. Another example is *BlenderBot* (Shuster et al., 2022b), where the LM decides to generate a query based on a prompt. In this case, the prompt corresponds to the instruction of calling the search engine tool. *BlenderBot* is capable of open-domain conversation, it has been deployed on a public website to further improve the model via continual learning with humans in the loop. Similarly, *ReAct* uses few-shot prompting to teach a LM how to use different tools such as `search` and `lookup` in Wikipedia, and `finish` to return the answer (Yao et al., 2022b). Similarly, Komeili et al. (2021); Shuster et al. (2022a) propose a model that learns to generate an internet search query based on the context, and then conditions on the search results to generate a response. *ReAct* interleaves reasoning and acting, allowing for greater synergy between the two and improved performance on both language and decision making tasks. *ReAct* performs well on a diverse set of language and decision making tasks such as question answering, fact verification, or web and home navigation.

In general, reasoning can improve decision making by making better inferences and predictions, while the ability to use external tools can improve reasoning by gathering additional information from knowledge bases or environments.

### 3.2.3   Searching and navigating the web

It is also possible to train agents that can navigate the open-ended internet in pursuit of specified goals such as searching information or buying items. For example, *WebGPT* (Nakano et al., 2021) is a LM-based agent which can interact with a custom text-based web-browsing environment in order to answer long-form questions. In contrast with other models that only learn how to query retrievers or search engines like *LaMDA* (Thoppilan et al., 2022) or *BlenderBot* (Shuster et al., 2022b), *WebGPT* learns to interact with a web-browser, which allows it to further refine the initial query or perform additional actions based on its interactions with the tool. More specifically, *WebGPT* can `search` the internet, `navigate` webpages, `follow` links, and `cite` sources (see Table 3 for the full list of available actions). By accessing the internet, the agent is able to enhance its question-answering abilities, even surpassing those of humans as determined by human evaluators. The best model is obtained by fine-tuning *GPT3* on human demonstrations, and then performing rejection sampling against a reward model trained to predict human preferences. Similarly, WebShop (Yao et al., 2022a) is a simulated e-commerce website where an agent has to find, customize, and purchase a product according to a given instruction. To accomplish this, the agent must understand and reason about noisy text, follow complex instructions, reformulate queries, navigate different types of webpages, take actions to collect additional information when needed, and make strategic decisions to achieve its goals. Both the observations and the actions are expressed in natural language, making the environment well-suited for LM-based agents. The agent consists of a LM fine-tuned with behavior cloning of human demonstrations (*i.e.*, question-human demonstration pairs) and reinforcement learning using a hard-coded reward function

that verifies whether the purchased item matches the given description. While there are other works on web navigation and computer-control, most of them assume the typical human interface, that takes as input images of a computer screen and output keyboard commands in order to solve digital tasks (Shi et al., 2017; Gur et al., 2019; 2021; Toyama et al., 2021; Humphreys et al., 2022; Gur et al., 2022). Since our survey focuses on LM-based agents, we will not discuss these works in detail.

## 3.3 Computing via Symbolic Modules and Code Interpreters

Although recent LMs are able to correctly decompose many problems, they are still prone to errors when dealing with large numbers or performing complex arithmetics (Mishra et al., 2022b). For example, vanilla *GPT3* cannot perform out-of-distribution addition, *i.e.* addition on larger numbers than those seen during the training even when provided with examples with annotated steps (Qian et al., 2022). In the context of reinforcement learning, the action space of a transformer agent is equipped with symbolic modules to perform *e.g.* arithmetic or navigation in Wang et al. (2022b). *Mind's Eye* (Liu et al., 2022b) invokes a `physics engine` to ground LMs physical reasoning. More precisely, a text-to-code LM is used to produce rendering code for the physics engine. The outcome of the simulation that is relevant to answer the question is then appended in natural language form to the LM prompt. As a result, *Mind's Eye* is able to outperform the largest LMs on some specific physical reasoning tasks while having two order of magnitude less parameters.

For reasoning graph generation, *CoCoGen* (Madaan et al., 2022) propose to generate python code generating a graph instead of a serialized version of the graph. *PAL* (Gao et al., 2022) relies on CoT prompting of large LMs to decompose symbolic reasoning, mathematical reasoning, or algorithmic tasks into intermediate steps along with python code for each step (see Figure 6). The python steps are then offloaded to a `python interpreter` outputting the final result. They outperform CoT prompting on several benchmarks, especially on GSM-HARD, a version of GSM8K with larger numbers. See Table 1 for a comparison between *PAL* and other models on GSM8K. Similarly, Drori et al. (2022); Chen et al. (2022b) prompts *Codex* (Chen et al., 2021) to generate executable code-based solutions to university-level problems, math word problems, or financial QA. For code generation, Shi et al. (2022) execute several sampled generation on a small number of test inputs, and use the output to select a solution. Instead, Mishra et al. (2022a) extends existing datasets with solutions written as python programs. Then, they finetune a model to generate python solutions to mathematical reasoning problems. In the context of theorem proving, Wu et al. (2022c) uses large LMs to automatically formalize informal mathematical competition problem statements in Isabelle or HOL. Jiang et al. (2022) generate formal proof sketches, which are then fed to a prover.

## 3.4 Acting on the virtual and physical world

While the previous tools gather external information in order to improve the LM's predictions or performance on a given task, other tools allow the LM to act on the virtual or physical world. In order to do this, the LM needs to ground itself in the real-world by learning about affordances i.e. what actions are possible in a given state, and their effect on the world.

**Controlling Virtual Agents.** Recent works demonstrated the ability of LMs to control virtual agents in simulated 2D and 3D environments by outputting functions which can then be executed by computers in the corresponding environment, be it a simulation or the real-world. For example, Li et al. (2022b) fine-tune a pre-trained *GPT2* (Radford et al., 2019) on sequential decision-making problems by representing the goals and observations as a sequence of embeddings and predicting the next action. This framework enables strong combinatorial generalization across different domains including a simulated household environment. This suggests that LMs can produce representations that are useful for modeling not only language but also sequential goals and plans, so that they can improve learning and generalization on tasks that go beyond language processing. Similarly, Huang et al. (2022a) investigate whether it is possible to use the world knowledge captured by LMs to take specific actions in response to high-level tasks written in natural language such as "make breakfast". This work was the first to demonstrate that if the LM is large enough and correctly prompted, it can break down high-level tasks into a series of simple commands without additional training. However, the agent has access to a predetermined set of actions, so not all natural language commands can be executed in the environment. To address this issue, the authors propose to map the commands suggested by

> **Question:** Roger has 5 tennis balls. He buys 2 more cans of tennis balls. Each can has 3 tennis balls. How many tennis balls does he have now?
> **Answer:** Roger started with 5 balls.
>
> ```
> tennis_balls = 5
> ```
>
> 2 cans of 3 tennis balls each is
>
> ```
> bought_balls = 2 * 3
> ```
>
> tennis balls.   The answer is
>
> ```
> answer = tennis_balls + bought_balls
> ```
>
> **Question:** The cafeteria had 23 apples. If they used 20 to make lunch and bought 6 more, how many apples do they have?
> **Answer:**
> **<LM>**

Figure 6: An example of few-shot PAL (Gao et al., 2022) prompt. **<LM>** denotes call to the LM with the above prompt. The prompts are based on the chain-of-thoughts prompting shown on Figure 1, and the parts taken from it are highlighted in green . In PAL, the prompts also contain `executable python code`, which performs operations and stores the results in the `answer` variable. When prompted with a new question, PAL generates a mix of executable code and explanation. The answer is obtained by executing the code and `print(answer)`.

the LM into feasible actions for the agent using the cosine similarity function. The approach is evaluated in a virtual household environment and displays an improvement in the ability to execute tasks compared to using the plans generated by the LM without the additional mapping. While these works have demonstrated the usefulness of LMs for controlling virtual robots, the following paragraph cover works on physical robots. Zeng et al. (2022) combine a LM with a visual-language model (VLM) and a pre-trained language-conditioned policy for controlling a simulated robotic arm. The LM is used as a multi-step planner to break down a high-level task into subgoals, while the VLM is used to describe the objects in the scene. Both are passed to the policy which then executes actions according to the specified goal and observed state of the world. Dasgupta et al. (2023) use 7B and 70B *Chinchilla* as planners for an agent that acts and observes the result in a PycoLab environment. Additionally, a reporter module converts actions and observations from pixel to text space. Finally, the agent in Carta et al. (2023) uses a LM to generate action policies for text-based tasks. Interactively learning via online RL allows to ground the LM internal representations to the environment, thus partly departing from the knowledge about statistical surface structure of text that was acquired during pre-training.

| Command | Effect |
|---|---|
| `search <query>` | Send <query> to the Bing API and display a search results page |
| `clicked on link <link ID>` | Follow the link with the given ID to a new page |
| `find in page:  <text>` | Find the next occurrence of <text> and scroll to it |
| `quote:  <text>` | If <text> is found in the current page, add it as a reference |
| `scrolled down <1, 2, 3>` | Scroll down a number of times |
| `scrolled up <1, 2, 3>` | Scroll up a number of times |
| `Top` | Scroll to the top of the page |
| `back` | Go to the previous page |
| `end:  answer` | End browsing and move to answering phase |
| `end:  <nonsense, controversial>` | End browsing and skip answering phase |

Table 3: The actions *WebGPT* can perform, taken from  Nakano et al. (2021).

**Controlling Physical Robots.** Liang et al. (2022) use a LM to write robot policy code given natural language commands by prompting the model with a few demonstrations. By combining classic logic structures and referencing external libraries, e.g., for arithmetic operations, LMs can create policies that exhibit spatial-geometric reasoning, generalize to new instructions, and provide precise values for ambiguous descriptions. The effectiveness of the approach is demonstrated on multiple real robot platforms. LMs encode common sense knowledge about the world which can be useful in getting robots to follow complex high-level instructions expressed in natural language. However, they lack contextual grounding which makes it difficult to use them for decision making in the real-world since they do not know what actions are feasible in a particular situation. To mitigate this problem, Ahn et al. (2022) propose to teach the robot a number of low-level skills (such as "find a sponge", "pick up the apple", "go to the kitchen") and learn to predict how feasible they are at any given state. Then, the LM can be used to split complex high-level instructions into simpler subgoals from the robot's repertoire. The LM can then select the most valuable yet feasible skills for the robot to perform. This way, the robot can use its physical abilities to carry out the LM's instructions, while the LM provides semantic knowledge about the task. The authors test their approach, called *SayCan*, on various real-world tasks and find that it can successfully complete long, abstract instructions in a variety of environments. To address the grounding problem, Chen et al. (2022a) propose *NLMap-SayCan*, a framework to gather and integrate contextual information into LM planners. *NLMap* uses a Visual Language Model (VLM) to create an open-vocabulary queryable scene representation before generating a context-conditioned plan. An alternative way of incorporating contextual information into the agent's decisions is to utilize linguistic feedback from the environment such as success detection, object recognition, scene description, or human interaction (Huang et al., 2022b). This results in improved performance on robotic control tasks such as table top rearrangement and mobile manipulation in a real kitchen. Finally, *RT-1* (Brohan et al., 2022) leverages large-scale, diverse, task-agnostic robotic datasets to learn a model that can follow over 700 natural language instructions, as well as generalize to new tasks, environments, and objects. *RT-1* makes use of *DIAL* (Xiao et al., 2022), an approach for automatically labeling robot demonstrations with linguistic labels via the vision-language alignment model *CLIP* (Radford et al., 2019).

## 4 Learning to reason, use tools, and act

The previous sections reviewed *what* LMs can be augmented with in order to endow them with reasoning and tools. We will now present approaches on *how* to teach them such abilities.

### 4.1 Supervision

A straightforward way of teaching LMs both to reason and to act consists in providing them with human-written demonstrations of the desired behaviours. Common ways of doing so are (i) via few-shot prompting as first suggested by Brown et al. (2020), where the LM is provided a few examples as additional context during inference, but no parameter updates are performed, or (ii) via regular gradient-based learning. Typically, supervised learning is done *after* an initial pre-training with a language modeling objective (Ouyang et al., 2022; Chung et al., 2022); an exception to this is recent work by Taylor et al. (2022), who propose to mix pre-training texts with human-annotated examples containing some form of explicit reasoning, marked with a special token. Some authors use supervised fine-tuning as an intermediate step, followed by reinforcement learning from human feedback (Nakano et al., 2021; Ouyang et al., 2022); see Section 4.2 for an in-depth discussion of such methods.

**Few-shot prompting.** Providing LMs with a few human-written *in-context* demonstrations of a desired behaviour is a common approach both for teaching them to reason (Wei et al., 2022c;b; Suzgun et al., 2022; Press et al., 2022) and for teaching them to use tools and act (Gao et al., 2022; Lazaridou et al., 2022; Yao et al., 2022b). This is mainly due to its ease of use: few-shot prompting only requires a handful of manually labeled examples and enables very fast experimentation as no model fine-tuning is required; moreover, it enables reusing the very same model for different reasoning tasks and tools, just by changing the provided prompt (Brown et al., 2020; Wei et al., 2022c). On the other hand, the ability to perform reasoning with chain-of-thoughts from a few in-context examples only emerges as models reach a certain size (Wei et al., 2022b; Chung et al., 2022), and performance depends heavily on the format in which examples are presented

(Jiang et al., 2020; Min et al., 2022), the choice of few-shot examples, and the order in which they are presented (Kumar and Talukdar, 2021; Lu et al., 2022b; Zhou et al., 2022). Another issue is that the amount of supervision that can be provided is limited by the number of examples that fit into the LM's context window; this is especially relevant if (i) a new behaviour is so difficult to learn that it requires more than a handful of examples, or (ii) we have a large space of possible actions that we want a model to learn. Beyond that, as no weight updates are performed, the LM's reasoning and acting abilities are tied entirely to the provided prompt; removing it also removes these abilities.

**Fine-tuning.** As an alternative to few-shot prompting, the reasoning and acting abilities of a pre-trained LM can also be elicited by updating its parameters with standard supervised learning. This approach has been used both for teaching models to use tools, including search engines (Komeili et al., 2021; Shuster et al., 2022b), web browsers (Nakano et al., 2021), calculators and translation systems (Thoppilan et al., 2022), and for improving reasoning abilities (Chung et al., 2022). For the latter, examples of reasoning are typically used in the larger context of *instruction tuning* (Mishra et al., 2021; Sanh et al., 2022; Wang et al., 2022d; Ouyang et al., 2022), where, more generally, an LM's ability to follow instructions is improved based on human-labeled examples. Examples are typically collected from crowd workers. In some cases, they can instead be obtained automatically: Nye et al. (2021) use execution traces as a form of supervision for reasoning, while Andor et al. (2019) use heuristics to collect supervised data for teaching a language model to use a calculator.

**Prompt pre-training.** A potential risk of finetuning *after* the pre-training phase is that the LM might deviate far from the original distribution and overfit the distribution of the examples provided during fine-tuning. To alleviate this issue, Taylor et al. (2022) propose to mix pre-training data with labeled demonstrations of reasoning, similar to how earlier work mixes pre-training data with examples from various downstream tasks (Raffel et al., 2020); however, the exact gains from this mixing, compared to having a separate fine-tuning stage, have not yet been empirically studied. With a similar goal in mind, Ouyang et al. (2022) and Iyer et al. (2022) include examples from pre-training during the fine-tuning stage.

**Bootstrapping.** As an alternative to standard fine-tuning, several authors propose to use *bootstrapping* techniques (e.g. Yarowsky, 1995; Brin, 1999) to leverage some form of indirect supervision. This typically works by prompting a LM to reason or act in a few-shot setup followed by a final prediction; examples for which the actions or reasoning steps performed did *not* lead to a correct final prediction are then discarded. For example, STaR (Zelikman et al., 2022) prompts a model to generate chain-of-thought reasoning sequences in a common sense question answering setup, but only keeps those chains that lead to the correct final answer for a given question. Finally, either the original LM or another (typically smaller) model is fine-tuned on all correct examples. As such, bootstrapping combines the data efficiency of few-shot prompting with some of the advantages of fine-tuning and can be successfully applied both to teach models to reason (Shridhar et al., 2022) and to use tools (Parisi et al., 2022).

## 4.2 Reinforcement learning

Supervised learning from human-created prompts is effective to teach models to reason and act. However, such data is difficult and costly to obtain. Human preference data — such as rankings or likes/dislikes — is much easier, faster, and cheaper to obtain than full demonstrations. For instance, it might be easier for a human to evaluate the quality of a summary than write one from scratch. Going further, it might be even easier to rank different summaries than scoring each separately. Such data cannot be used in a supervised setting, but can provide rewards in the context of Reinforcement Learning (RL) (Sutton and Barto, 2018).

RL has proven successful for learning complex behaviors through feedback-based interaction with an environment, and it has been used for applications such as playing games (Mnih et al., 2015; Silver et al., 2016; Vinyals et al., 2019; Team et al., 2021; Bakhtin et al., 2022) or controlling robots (Gu et al., 2017; Kalashnikov et al., 2018; Akkaya et al., 2019; Lee et al., 2020). When training a LM with RL, the LM can be considered an agent that learns a policy (i.e. a distribution over the model's vocabulary from which the next token is sampled) in order to optimize some reward function. Most of the existing work on RL and ALMs has focused on teaching LMs how to act rather than reason. The closest work on learning how to reason via RL is STaR (Zelikman et al., 2022), a bootstrapping-based approach that is discussed in Section 4.1

RL is a natural framework for training LMs to act and use tools since many of these tools are non-differentiable (e.g. search engines, calculators or programming language interpreters). Additionally, many tasks that benefit from interacting with tools resemble sequential decision making problems (e.g., navigating a web-browser to buy a specified product) and have a well-defined reward (e.g., 1 if the model buys the correct product and 0 otherwise). While there are early works focused on models that could interface with external tools, they employ ad-hoc tool-dependent architectures (Adolphs et al., 2022; Buck et al., 2018; Nogueira and Cho, 2017; Zhong et al., 2018). We do not cover them here since the main focus of our survey is instead on the acting and reasoning capabilities of standard general-purpose LM architectures trained with the language modeling objective.

**Hard-coded reward functions.** When teaching a LM how to use external tools, the standard practice is to update the weights of the model using a scalar reward generated by a hard-coded reward function. This task-dependent function is computed based on the tool output. The LM agent takes a textual input, which in RL terminology corresponds to the current state of the environment, and generates a sequence of tokens, or actions in RL terms. Optimization is done through policy gradient algorithms like REINFORCE (Williams, 1992), PPO and similar variants (Schulman et al., 2017; Ramamurthy et al., 2022).

Initial works on training LMs to use tools via RL mostly focused on searching and fetching additional factual information. Common tools for such information-seeking tasks are `document retrievers`, `question answering systems`, and `search engines`. The first two consist in retrieving document from a pre-defined set of text documents, or in retrieving an answer based on some input query. However, a search engine allows for more structured interactive search where, for instance, the model further refines the initial query or performs additional actions based on the initial output of the tool. For example, Wu et al. (2022d) perform conversational question-answering by teaching a LM via RL to rewrite queries in order to feed them to an off-the-shelf retriever. The reward function is a contrastive retrieval-accuracy metric based on the token overlap between following conversation rounds and retrieved passages. Another example is the work from Liu et al. (2022a): *RAINIER* is a LM able to generate contextually relevant questions that are optimized to query a frozen `QA system`. After distilling knowledge from a larger *GPT3* (Brown et al., 2020) model into a smaller *T5* model (Raffel et al., 2020), *RAINIER* is finetuned using PPO (Schulman et al., 2017) with feedback provided by the pre-trained question answering model from Khashabi et al. (2020). Interestingly, this work is an example of a LM learning to use another frozen neural model as an external tool.

Yao et al. (2022a) use RL to teach a language model to navigate a `virtual shop` and buy items constrained on attributes like color and price. Similar to *WebGPT* (Nakano et al., 2021), the model is given a goal in textual format and allowed to perform a limited set of actions. Prompted with a user-generated instruction, in a multi-task learning setup, the model needs to simultaneously understand the query and browse the web to search for the right product. The reward is a hard-coded text-matching function based on the similarity between the model-purchased written description of the item and the given shopping instruction. Optimization is performed with the A3C algorithm (Mnih et al., 2016), a variant of the standard actor-critic method. While the model still lags behind human experts, they found that fine-tuning with RL after training on human demonstrations improves performance. This provides additional evidence of the benefits of reward-based learning for endowing LMs with the ability to interact with external tools.

While interacting with a `search engine` or a `document retriever` allows a model to augment its current context with additional input, it is often necessary to process structured information when interacting with tools like a `knowledge base`. Dognin et al. (2021) train a LM to learn how to interface with a graph-based knowledge base by performing the text2graph and graph2text tasks. The model, based on a *T5* architecture (Raffel et al., 2020) and trained with the vanilla policy gradient algorithm REINFORCE (Williams, 1992), can perform bidirectional generation of text and graphs and shows state-of-the-art performance on tasks related to knowledge base automated construction from text and vice versa. The *T5*-based agent is trained to directly maximize graph2text metrics such as BLEU (Papineni et al., 2002a), METEOR (Banerjee and Lavie, 2005), and chrF++ (Popović, 2017), or text2graph ones such as F1, Precision, and Recall.

Finally, it is also possible to leverage RL at inference time only. For example, Cao et al. (2023) propose a method to avoid generating tokens that will likely lead to toxic content by framing it as a RL problem. More precisely, a reward model and a value function are trained to evaluate respectively the toxicity of a sentence

and the probability of the so-far generated tokens to lead to a toxic generation. At inference time, the latter is used to truncate the next token probability distribution towards the tokens that are less likely to lead to a toxic generation.

**Human feedback.** Evaluating the quality of machine-generated text is non-trivial because it can vary depending on the context, individual preferences, and user's intentions. For example, in some contexts, a user might require creative writing, while in others it may just require factual information. Model outputs should be judged accordingly and should be able to capture such intent differences. Several metrics based on heuristics like BLEU (Papineni et al., 2002b) and ROUGE (Lin, 2004) have been developed for comparing model outputs to reference texts. However, they fail to fully capture the quality of generations with respect to human intentions. Human feedback can be exploited to improve the quality of machine-generated text, for example for dialog agents (Xu et al., 2022). In particular, Reinforcement Learning from Human Feedback (RLHF) (Knox and Stone, 2008; MacGlashan et al., 2017; Christiano et al., 2017; Warnell et al., 2018) aims to overcome these limitations by using human preferences as an evaluation metric and as an objective function to optimize the language model. Using RLHF allows LMs to be more closely aligned with complex human preferences and values which are difficult to capture by hard-coded reward functions.

RLHF works by using a pre-trained LM to generate text, which is then evaluated by humans by, for example, ranking two model generations for the same prompt. This data is then collected to learn a reward model that predicts a scalar reward given any generated text. The reward captures human preferences when judging model output. Finally, the LM is optimized against such reward model using RL policy gradient algorithms like PPO (Schulman et al., 2017). RLHF can be applied directly on top of a general-purpose LM pre-trained via self-supervised learning. However, for more complex tasks, the model's generations may not be good enough. In such cases, RLHF is typically applied after an initial supervised fine-tuning phase using a small number of expert demonstrations for the corresponding downstream task (Ramamurthy et al., 2022; Ouyang et al., 2022; Stiennon et al., 2020).

A successful example of RLHF used to teach a LM to use an external tool stems from *WebGPT* (Nakano et al., 2021), discussed in 3.2.3, a model capable of answering questions using a `search engine` and providing references to support such answers. The tool interface is a simplified text-based web-browser. The model architecture is based on *GPT3* (Brown et al., 2020) and is trained to perform browsing actions expressed in natural language. The model is fine-tuned on question-human demonstration pairs, before further optimization via RLHF. On two QA datasets, *WebGPT*'s answers are preferred relative to human-generated ones and tend to be more factual than the original vanilla *GPT3* model. Similarly, Menick et al. (2022) propose *GopherCite*, a *Gopher*-based LM model (Rae et al., 2021) fine-tuned with RLHF that can cite supporting evidence when answering questions and abstain from answering when unsure. In contrast with *WebGPT*, *GopherCite* uses an information retrieval external module rather than a web-browser to find relevant information that improves its question answering capabilities. Besides learning to use external tools, RLHF has also proven useful for a wide range of language generation tasks, from summarization (Ziegler et al., 2019; Wu et al., 2021; Stiennon et al., 2020) to training more helpful, harmless, and accurate assistants (Glaese et al., 2022; Cohen et al., 2022; Ouyang et al., 2022; Bai et al., 2022). Since these works do not focus on training models to reason and act, they are out of the scope of this survey.

### 4.3 Limitations and future directions

Despite recent algorithmic progress and performance improvements, current RL methods still suffer from instability issues which can make training difficult and slow (Ramamurthy et al., 2022; Snell et al., 2022). While supervised learning has been an efficient and robust way to fine-tune language models on specific tasks (Mishra et al., 2021; Sanh et al., 2022; Wang et al., 2022b), this assumes the existence of a large number of expert demonstrations, which can be difficult and costly to obtain. This is particularly true for tasks that require reasoning and acting where we do not have readily available data. A possible solution to the lack of quality data problem could come from bootstrapping methods and offline RL. The promise of these methods is that a dataset can be generated via feedback and interactions of any behavior policy and can be used to train an improved policy. Therefore, combining a data-driven approach with offline RL could give the "best of both worlds": learning from counterfactual events in a scalable and stable

way (Levine et al., 2020). In the offline regime, though, there are some open challenges. The maximum improvement can be limited by a number of factors such as: the suboptimality of the initial behavior policy, the dimensionality of the state and the action space, the length of the effective horizon, the accumulation of errors and distributional shifts due to the discrepancy between the behavior policy and the learned one (Levine et al., 2020). Recent works (Zelikman et al., 2022; Snell et al., 2022) have shown that such approaches could reach performance that goes beyond that of the expert demonstrations or improve over initial model generations. For example, Snell et al. (2022) introduce a new offline RL algorithm called ILQL which learns from a static dataset of demonstrations and their associated rewards by estimating a value function and using it to optimize LM generations. ILQL combines online RL flexible optimization framework with the simplicity and ability to learn from existing datasets of supervised learning, resulting in good performance on dialogue tasks. As explained in Section 4, Zelikman et al. (2022) employ a bootstrapping approach for teaching LMs to reason, which can be seen as an approximation to policy gradient algorithms.

Recently, Schick et al. (2023) proposed *Toolformer*, a model that teaches itself to use tools in a self-supervised way. This is achieved by first using the few-shot abilities of an existing LM to sample a large amount of potential tool uses. For instance, the model can call a calculator API to augment its context, e.g., "*Out of 1400 participants, 400 (or [Calculator(400 / 1400)→ 0.29] 29% passed the test.*" Then, the model is fine-tuned on its own generations, filtering them based on whether they reduce perplexity for future tokens generations. This method enables using several tools (e.g., a `calendar`, a `calculator`, or an `information retrieval system`). However, it was tested in a limited setup of using a single tool at once, since examples of tool use were independently sampled. We believe that studying how this approach could be extended to more complex multi-step tool uses is a promising research direction for a generalist LM-based agent.

## 5 Discussion

**Moving away from language modeling.** Is a model trained to do intermediate reasoning steps or having access to the internet still purely performing language modeling? Indeed, in NLP, language modeling (Bahl et al., 1983) is generally defined as the task of predicting missing tokens given a context and is relied heavily on for pre-training models. However, several techniques have been developed to later fine-tune models (Ziegler et al., 2019; Wei et al., 2022a; Sanh et al., 2022) to perform various natural language tasks, which could be seen as moving away from traditional language modeling. In particular, the texts used to fine-tune LMs are not just found on the internet, but rather designed to explicitly inject some level of grounding. One of the argument advocated recently in Goldberg (2023) is that "*it might be much easier to learn from direct instructions like these than it is to learn from non-instruction data*". This argument can be supported by the recent work of Giannou et al. (2023), showing both theoretically and in practice that even shallow looped transformers can follow instructions and be programmed as general purpose computers. Intuitively, a text is the result of complex intermediate thoughts that are hidden. Therefore, the superficial text used for supervision can be seen as representing only the logs of these thoughts, thus lacking of context. Conversely, with task-oriented supervised data, we can explicitly ground the answer with the intermediate steps. In this regard, the resulting model may not be considered as a language model. And yet, the task is still about predicting the next token given text only. The argument is all the more true for ALMs since they can augment their context.

In particular, tool-augmented LMs might actually lose the ability to assign a probability to the next token - which is at the core of language modeling: whereas a regular LM can easily compute $p(x_t \mid x_1, \ldots, x_{t-1})$, a tool-augmented LM has to consider all possible tool uses, e.g. $p(x_t \mid x_1, \ldots, x_{t-1}) = \sum_c p(c \mid x_1, \ldots, x_{t-1}) \cdot p(x_t \mid x_1, \ldots, x_{t-1}, c)$ where $c$ is a tool, which might not be tractable. First, in the general case, marginalizing over all the tools is not enough, one also has to marginalize over all possible tool use. For many tools however, the probability $p(c \mid x_1, \ldots, x_{t-1})$ associated with each possible tool output cannot be modeled. Take for example web browsing: for a fixed query, the result may vary day to day unpredictably as both the internet and the search algorithms are constantly evolving, making the evaluation of $p(c \mid x_1, \ldots, x_{t-1})$ impossible unless the tool uses a past snapshot of the internet, which is generally not wanted. Second, in the case where a tool can be called within another tool call, the number of tool uses can become exponential in the number of tools. For example, within a tool call, an ALM could browse the web by reading the content of a web page,

which is text, then apply another of the tools to the text it found to refine its query, etc. If for some reason we require a depth that is equal to the number of tools, we have an exponential complexity.

For these reasons, we refer to Augmented Language Models (ALMs) in this survey, to distinguish from Language Modeling in the traditional sense.

**A tradeoff between memorizing and querying tools.** Is it preferable to memorize information in the model weights, or to leverage external tools? Some situations arguably require external tools, for example computing $213443^{344}$. However, many information are well known facts such as "The Eiffel tower is located in Paris" or $1 + 2 = 3$, and should not be offloaded. And, when learning world representations, memorization is not only desirable, but also deeply connected to reasoning (Hayes et al., 2014). Can ALMs be calibrated enough to decide when and when not to use a tool? Could a computation budget for each tool be integrated into the loss to let the model learn to do so?

**Generalizing the non-parametric framework.** A motivation behind information retrieval augmented LMs such as *RETRO* (Borgeaud et al., 2022) and *Atlas* (Izacard et al., 2022) is to develop a class of LM requiring less parameters through relying on an external non-parametric memory. The motivation for using other kind of tools such as `code interpreter` or `calculator` has been slightly different so far: for instance, Cobbe et al. (2021) use a calculator to improve accuracy on tasks requiring arithmetic. Yet, the paradigm of tool-augmented LMs can be seen as a generalization of the non-parametric framework. Indeed, beyond information retrieval, LMs can delegate any kind of abilities such as calculus to the corresponding external tools. By avoiding to store rarely accessed knowledge in their weights, tool-augmented LMs may have better scaling laws and thus yield smaller models retaining the capabilities of their largest counterpart. Combined with the possibility to access recent information from the external world thus avoiding frequent updates, non-parametric generalization holds great benefits for ALMs.

**A path towards autonomous machine intelligence?** A concept for an autonomous intelligent agent was proposed by LeCun (2022). We now discuss to what extent ALMs instantiate this idea. In LeCun (2022), the agent is composed of different modules starting from a world model and a short-term memory. Essentially, the agent takes actions via an actor module based on its world model, perception module, and short-term memory so as to minimize some cost. The agent is also equipped with a configurator module for modulating the world model, the perception, the actor and the cost given the task at hand.

Translating into this framework, the ALM's weights essentially contain the world model, perception and actor modules. The short-term memory can be identified with the ALM's context or prompt. Based on its perception of the context and its world model, the ALM would take actions by outputting special tokens, and perceive the result. The configurator module remains elusive but may be implicit: it can be seen as the conditioning induced by the ALM's context, for example an initial prompt such as "You are a kind and helpful assistant". Finally, the cost remains fixed in this framework, and could be the ALM's perplexity mixed with a computational cost associated to reasoning and using external tools.

However, an important feature of the agent in LeCun (2022) is its ability to plan, defined by the decomposition of a complex task into subtasks: in the ALM's context, planning is akin to reasoning, a slight abuse of terminology as it is not clear whether LMs reason as humans do as noted in Section 2. LeCun (2022) propose to implement reasoning (under the term planning) as the minimization of an energy with respect to a hierarchical combination of actions. Since ALMs only perform predictions at the token level, they cannot reason according to LeCun (2022)'s view and may be still limited to System 1 tasks, *i.e.* that rely on reflex rather than logic and thinking. Whether System 2, *i.e.* the opposite abilities can be obtained by pushing current methods remains uncertain. For example, LMs are deprived from global consistency beyond their maximum sequence length: as an illustration, two different discussions with the same LM will result in inconsistencies. This is a strong limitation when it comes to solving complex problems that require to perform a large number of sub-goals such as writing a research paper, where one has an initial mental state that includes the current results and the angle of the paper. This process is not linear and results from different interactions, e.g., new ideas while reading some related works. The mental state is maintained although updated trough all the process, such that we keep in mind the big picture. Although more compute and

larger input size could mitigate the issue, another solution may be to endow LMs with adequate components. In this regard, a model architecture that intrinsically makes the LM consistent with an energy function as suggested in LeCun (2022) could constitute a promising venue.

Finally, our survey sees LMs as the central piece of a generalist agent that could reason in natural language and interact with external tools. Along these lines, Wang et al. (2023) uses a LM as a centralized planner to generate goal sequences for solving tasks in the game of Minecraft. Through a feedback loop and intermediate checks on subgoals execution, the LM can explain mistakes of the goal executor and refine its original plan. However, we note that a LM-based controller might not be the only viable approach for a generalist agent. Recent work on the game of Diplomacy (Bakhtin et al., 2022), a long-standing challenge for AI agents due to its complex planning and reasoning dynamics, employs an ad-hoc planning model trained via self-play and reinforcement learning. Here the LM is used to interact with other players, thus as an external communication module grounded in the current state of the game. This offers an alternative view of LMs as agents specialized to communicate with humans, albeit in the restricted setting of a Diplomacy game. We believe that (A)LMs will play a central role in the next generation of powerful interactive systems, whether as centralized controller of a modular system or as a language-only module that needs to interact with an orchestrator remains an open research question.

**Augmented Language Models benefits.** Overall, ALMs offer many potential advantages over traditional LMs.

- *Truthfulness*: As the current LM's training objective is arguably responsible for inciting the generation of seemingly plausible but not factual information, grounding the predictions through some tools should lead to more trustworthy models. However, although this conclusion is straightforward when equipping a LM with a calculator, there is surprisingly little evidence of it for information retrieval augmented LMs (Krishna et al., 2021). One of the reasons is the presence of a lot of non-truthful information in the web. Investigating this direction will be critical for making LM reliable.

- *Estimating and reducing uncertainty*: Extending the maximum-likelihood paradigm by letting the model reason and access additional information could help models to learn what they know and what they don't. Some papers suggest that LMs are already well calibrated (Kadavath et al., 2022), i.e. there is a high correlation between the accuracy of their predictions and the corresponding likelihood. This uncertainty could be directly exploited by ALMs to know when to rely on their own weights, or when to query an external tool.

- *Interpretability*: Deep learning models are often considered to be black boxes, and their predictions are difficult to interpret. Providing intermediate reasoning steps and relying on tools should help to make ALMs more interpretable. In particular, we can expect that being able to cite the sources used to compose the answer to be critical. However, some works Lewkowycz et al. (2022) pointed out that chain-of-thoughts can lead to the correct predictions even though the intermediate reasoning doesn't make any sense, indicating clear challenges for researchers exploring this direction.

- *Enhanced capabilities*: ALMs with improved reasoning abilities and tools can be more helpful assistants and solve a wider range of tasks than standard LMs. For example, an ALM connected to a python interpreter can run code and experiments on a user's behalf, which a vanilla LM cannot do. In addition, a feedback loop can emerge between reasoning and acting, where each ability further improves the other (Yao et al., 2022b). Interacting with external tools, entities, and environments can improve reasoning since it allows the ALM to collect additional information and ground itself in the real-world. Similarly, reasoning can improve the ALM's decision making abilities such as when and how to use a certain tool.

**Cost of using tools.** To the best of our knowledge, the cost of using tools has not yet been taken into account comprehensively.

Overall, using tools seem to be beneficial in terms of energy use. Retrieval-augmented language models are a good example of a more efficient way to handle new information since updating an external data store can be

orders of magnitude cheaper than re-training a LLM for scratch every time we get new data (as argued for example in Borgeaud et al. (2022)). This stays true for using a calculator or browsing the web for example. One can therefore assume that relying on tools rather than storing knowledge in parameters is generally more energy-efficient than training, frequently updating a LLM, and scaling it if some ability remains out of reach, if one assumes an optimal tool usage (i.e., not using each tool at each token). Then, how to assess the cost of using a tool versus another?

We believe that the most natural way of estimating such cost is to consider two factors: (i) the price per tool request, and (ii) the time required to fulfill the request. These factors make sense from the perspective of the ALM, and are a reasonable way to take into account the cost of creating and maintaining the tool since this should typically be reflected in the pricing. Interestingly, including such cost at training and inference time under the form of a loss term for example could help ALMs to develop optimal tool use, and lead to the emergence of models that know how to balance between tool use or internal chain of thoughts.

**Ethical concerns.** ALMs raise new ethical concerns. LM predictions based on tools may look more trustworthy and authoritative at a first glance, when in fact many of them will still be incorrect. Moreover, we can expect this phenomenon to be amplified as LMs reason in quite a similar manner to humans (Dasgupta et al., 2022), making it even harder to detect mistakes. Hence, ALMs may be leveraged to generate more convincing fake news and conspiracy theories. Conversely, conditioning ALMs on "safe" and trusted sources should lead to more factuality and less toxicity in their answers.

While such ethical concerns apply to most tools, it is important to distinguish between passive and active tools. The former only collects external information and passes it to the LM's context, while the latter allows it to act on the virtual or physical world without human validation in the loop. An example of active tool is letting the LM control a search engine. Hence, there exists a broad spectrum of possible harmful consequences of LM usage. We are moving from passive LMs that generate text in isolation of the external environment, towards ALMs that act in the real world. In this context, the aforementioned ethical concerns may resonate even further, as ALM will be connected to more and more tools and environments.

Currently, training LLM results in a large number of gas emissions, leveraging tools may result in smaller hence cheaper models that are more environmentally friendly both at training, inference, and update. ALMs are therefore a promising avenue to decrease the environmental impact of LLMs.

ALMs also have the potential to transform the LM landscape. Nowadays, state of the art LLMs have been the product of large research organizations, fine-tuning these models towards more versatility (via reasoning and learning to use different tools) and agency (via actions) may be within range of smaller entities and perhaps individuals. We may therefore see the burgeoning of a market of open, possibly specialized and even personalized ALM agents, learning to use new tools, handling various data modalities and interacting with each other. We may also see the emergence of a new breed of app store, dedicated (via appropriate documentation or API constraints for example) to be used by ALMs. Overall, this new ecosystem may further accelerate the current pace of LLM research and application to the real world, while also making its control even more uncertain.

## 6  Conclusion

This survey presented works in which LMs are augmented with better reasoning and tools. In most of the works, LMs augment their context with additional relevant information useful to perform missing token prediction. As many of these augmentations are non-parametric, *i.e.* involve calling an external, possibly non-parametric module, such LMs arguably depart from the classical language modeling paradigm, hence our decision to dub them Augmented Language Models. Although several works focus either on reasoning or acting skills of LMs, most of them rely on human annotation which may not be scalable, e.g., hand-crafted few-shot prompting, or reinforcement learning from human feedback. How to equip language models with meaningful augmentations in a fully self-supervised fashion remains an open research question. Additionally, as very few works combine reasoning and tools, future efforts should study the integration and fruitful interaction between these two skills. Overall, we believe studying Augmented Language Models is a promising

and exciting research avenue towards the next generation of deep learning systems capable of complex and useful human-machine interaction.

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
