# OpenReview forum: "Augmented Language Models: a Survey"
_TMLR — Accepted by TMLR_

### Review · Reviewer_7WnD · 2023-04-30

**Summary Of Contributions:**

This paper provides a comprehensive review and survey of **augmented language models** (ALMs), broadly defined as language models that are augmented with reasoning skills and the ability to use tools. The paper begins with a summary of the shortcomings of current LMs, and **why** the shift towards ALMs may alleviate these issues: Concretely, current LMs often (i) suffer from hallucinations, (ii) require a large number of model parameters and dataset size to master emergent behaviour, and (iii) are not straightforward to update through continual learning.

The paper then dives deeper into two promising sub-areas that can address these limitations: (i) **reasoning** (e.g. prompting, chain-of-thought prompting, etc.), and (ii) **tool use and acting** (e.g. calling another model, using a calculator, search engines, retrieving information, taking actions in the real-world, etc.). In each section, the paper provides a comprehensive list and summary of the many different earlier work that had been done in each research area, alongside some of the current limitations of these approaches.

Lastly, the paper discusses **how** current ALM approaches teach the model to acquire the relevant ability or tool use, and concludes by discussing some of the broader implications (e.g. trade-off between storing the knowledge and ability in the model weights vs keeping them in the external tools, ethical concerns of ALMs, etc.).

**Audience:**

Yes

**Broader Impact Concerns:**

The paper includes an "Ethical Concerns" subsection in page 20. This subsection can be more comprehensive by touching on other relevant ethical issues. For instance, training a large dense LM on a large compute cluster can result in a large number of emissions; if leveraging tools can result in smaller models that are cheaper to train, deploy, and update, then this can be a good thing in terms of ethical concerns and environmental impact.

I am not familiar with this literature, but if there is prior work that assesses the safety / factuality / toxic language rate of tool-augmented vs non-tool-augmented standard LMs, it would be worth citing (or at the very least mentioned as an important open research question that the community should work on more in the near future). Also there is a possibility of dual use here: more capable ALMs that can e.g. leverage  and condition on more recent fake news and conspiracy theories on social media can then be used to generate more even more convincing, timely, and relevant-looking fake news that convinces people of things that are not true for e.g. one's political gains.

**Claims And Evidence:**

Yes

**Requested Changes:**

1. **Critical**: More discussion around the limitations and open questions around each subsection and more broadly. Examples of important topics that should be covered in more depth:
    - In terms of retrieval-augmented models, how do we do fast retrieval, especially when the retrieval set is huge (e.g. the whole internet)? Is exact inference tractable, and what approaches should we use when it is not? Can the retrieval be done on-device (e.g. GPUs), and what happens if it doesn't fit on the GPU memory?
    - Despite the benefits, what are the primary **costs** of doing ALMs? How do the computational costs of these approaches (e.g. calling a search engine or a calculator or keeping a large retrieval set) compare against trying to cram all the knowledge inside a single dense model? What are the different dimensions (e.g. compute, memory, costs of API calls to other tools) of these costs? Has prior work provided a thorough accounting of these costs? If not, how should we start to measure these?
    - How scalable are some of these approaches? For instance, if we have to marginalize over all the different tools at the LM pre-training time, it would likely be very expensive given that the number of tokens is very large. Can we do this as a "second-stage" pre-training on a smaller dataset instead?
    - A key part of tool use is knowing **when** to rely on the tools, and **which** tools to use for a given context. The discussion around this important topic seems sparse at the moment; what are some avenues that the community should explore more here?
    - The paper identifies combining reasoning and tool use as a promising avenue. This is worth discussing in more detail: Are there any obvious low-hanging fruits that we can try? What is the right "medium" for combining them: a lot of reasoning tools, theorem provers, etc. operate through formal languages, whereas LLMs operate mostly through natural language. Is this an important issue, and if so, how should we bridge them?
    - The paper mentions "offline RL" (page 18) as a potential solution; it might be worth discussing some of the pros and cons of offline RL, e.g. better safety through "sandboxing", although exploration is more difficult, etc.

2. **Critical**: In the "moving away from language modelling" subsection (page 18), the paper claims that "... where c is a tool, which might not be tractable". Three points:
    - I don't agree that this is necessarily intractable: The formal definition of tractability refers to problems that can be solved in polynomial time. Given a finite set of N tools, we simply need to consider and marginalize over all those N possibilities, which can still be done in polynomial time with respect to the number of tools N. It can be **expensive** when the number of tools N is large, but not intractable.
    - In terms of the equation, the paper states that $p(x_t \mid x_1, \cdots, x_{t-1}) = \sum_{c} p(c) * p(x_t \mid x_1, \cdots, x_{t-1}, c)$. Shouldn't $p(c)$ here be $p(c \mid x_1, \cdots, x_{t-1})$? I'd imagine the choice of the tool should depend on the context, and this formulation lends itself well to marginalization & latent variables (here the choice of c is treated as a latent variable that we simply marginalize over).
    - The paper alludes to ALMs being somewhat of a shift from the standard, probabilistic LM formulation. I don't agree that this is necessarily the case: With the formulation above, ALMs **still** define a probabilistic distribution over the next token (and through the chain rule, over the entire sequence, where $p(\mathbf{x}) = \prod_{x_t \in \mathbf{x}} p(x_t \mid x_{<t})$, only that now each prediction $p(x_t \mid x_{<t})$ involves a **marginalization** over all possible tools / retrieval documents, etc.). The fact that each prediction $p(x_t \mid x_{<t})$ is now **more expensive** due to the need to marginalize does **not** mean that it is any less probabilistic than the standard LM or that we have shifted away from the probabilistic interpretation of LMs, as both standard LMs and ALMs still define valid probability distributions over the next token. In practice, one can rely on approximate inference procedures (as opposed to exact inference when the number of tools N is large) to approximate exact marginalization, as the community has done a lot in the past when it comes to other types of latent variables.

3. **Recommended**: The definition of taking actions can perhaps be made more precise. In some sense, standard LMs also "take action" in the form of generating a sequence of words that will be displayed on the screen. I don't think this is what the paper means by taking action (as opposed to e.g. selecting a tool / reasoning step as a latent variable, or more actively controlling a web browser)?

4.  **Typo**: In page 14, "... gather an integrate contextual ..." is a typo, "an" -> "**and**"?

**Strengths And Weaknesses:**

**Strengths**

1. This paper provides a comprehensive overview of earlier work in the ALM literature. As someone who is broadly familiar with this literature, by reading this survey paper I nevertheless learn about interesting and relevant prior work that I had not encountered before, which attests to the comprehensive nature of the survey.

2. The ALM literature (both in terms of reasoning, tool use, and action-taking through LLMs) is incredibly broad and fast-paced, which presents a high barrier to entry for researchers who are new to this critical research area. Due to its comprehensive nature, this paper constitutes a great overview and entry point to this important research area; these readers can then dive deeper into their sub-area of choice by reading some of the cited papers that are most relevant to their interests.

3. The paper is overall well-structured and well-written, beginning with why ALMs are necessary, why the focus on reasoning and tool use, followed by a comprehensive description and current open questions of each research area, and concluding with a broader discussion of the rise of ALMs.

**Weaknesses**

1. While the survey paper is comprehensive in terms of the description of the relevant prior work, the "limitations" subsection of each section is much shorter. In my opinion, these limitations & open questions constitute an important part of a survey paper: Not just outlining what **has** been done, but also dedicating enough space to comprehensively outline **what is still missing and what we as a community should do more in the future**: (i) what are some of the open challenges (computational or otherwise) behind each approach? (ii) Are there any impediments to scaling the approaches, considering that scale is a key driving force behind recent progress? (iii) What are the pros and cons of the proposed approaches against other alternatives (e.g. online vs offline RL)? Etc. Some examples of what would be important to cover in more depth are provided in the "requested changes" section below.

2. I have some objections to some of the assertions in the paper, particularly regarding whether or not these semi-parametric forms of LMs still constitute the probabilistic language modelling objective (Section 5 "Discussion", "Moving away from language modelling", page 18). This is detailed in the "Requested Changes" section below.

---

> ### Author Response · Authors · 2023-05-25
> **1/2**
>
> We thank the reviewer for their positive feedback.
>
> - Our definition of taking actions for ALMs states “calling a tool having an effect on the virtual or physical world”. Since we indeed do not consider generating tokens to be an action in this context, we propose to replace this by “calling a tool that modifies a state in a virtual or physical object”.
> Fixed typo p14.
> - The ethical concern section has been strengthened (see general answer).
>
> **More discussion around the limitations and open questions**
>
> *Retrieval and efficiency*
>
> We agree with the reviewer that information retrieval step efficiency is an important question to address in order to enable large scale retrieval. In fact, there have been quite a few works on this topic, and we believe this is not a limitation anymore. We added an “efficient large-scale retrieval” subsection in the section 3.2 of the paper, that describes how these issues have been addressed:
>
> Efficient large scale retrieval. Since retrieval-augmented language models store knowledge in an external data store, it is crucial that the information retrieval step is efficient  when dealing with a large number of document/passage/sentence embeddings and/or a large number (billions) of query vectors. The literature offers a variety of optimisations to boost performance of retrievers as well as reducing the memory footprint of the data store.
>
> First, an index structure can be built for the document/passage/sentence embeddings and efficient similarity search can be performed, for instance using the Facebook AI Similarity Search library (faiss) (Johnson et al., 2021). Leveraging indexing structures enables efficient search operations by partitioning the vector space and enabling fast pruning of irrelevant vectors.
>
> Second, if there exist memory constraints due to a large number of document/passage/sentence embeddings, a multi-node, multi-gpu distributed framework can be used to enable storing the index corresponding to only part of the document/passage/sentence embeddings per process. The query embeddings can also be split in batches and obtain the top-k results in parallel using GPUs for further speed ups for approximate or exact search. When exact search becomes intractable due to scale, approximate nearest neighbor algorithms can be used. These algorithms trade off accuracy for speed, allowing significantly faster retrieval while maintaining reasonably accurate results, see Johnson et al. (2021) for further details.
>
> Finally, in order to further reduce the memory footprint of each of the index shards corresponding to a subset of the document/passage/sentence embeddings, different compression techniques can be used. For instance,the Atlas retrieval-augmented language model (Izacard et al. 2022) uses product quantisation (Jégou et al., 2011) to reduce the memory footprint of the retriever without sacrificing accuracy in downstream tasks in terms of exact match and recall@50 metrics of Q&A.
>
> Currently, there also exist a variety of vector database frameworks that have some of these optimisations
> implemented out of the box and can be leveraged to use retrieval as a service without requiring the user to implement all optimisations from scratch.
>
> *Cost of ALMs and tools.*
>
> This is a very relevant point. We added the following discussion to the survey:
>
> To the best of our knowledge, the cost of using tools has not yet been taken into account comprehensively.
>
> Retrieval-augmented language models are a good example of a more efficient way to handle new information since updating an external data store can be orders of magnitude cheaper than re-training a LLM for scratch every time we get new data (as argued for example in Retro). This stays true for using a calculator or browsing the web for example. One can therefore assume that relying on tools rather than storing knowledge in parameters is generally more energy-efficient than training, frequently updating a LLM, and scaling it if some ability remains out of reach, if one assumes an optimal tool usage (i.e., not using each tool at each token). Then, how to assess the cost of using a tool vs. another?
>
>
> We believe that the most natural way of estimating such cost is to consider two factors: (i) the price per request, and (ii) the time required to fulfill the request. These factors make sense from the perspective of the ALM, and are a reasonable way to take into account the cost of creating and maintaining the tool since this should typically be reflected in the pricing. Interestingly, including such cost at training and inference time under the form of a loss term for example could help ALMs to develop optimal tool use, and lead to the emergence of models that know how to balance between tool use or internal chain of thoughts.

---

> > ### Author Response · Authors · 2023-05-25
> > **2/2**
> >
> > *Scalability: When to rely on tools, and which tools to use.*
> >
> > Knowing when to call a tool and which is crucial for ALMs scalability (versus using each tool for each token). Although this is currently an open question, there are different but equally promising avenues to know when to rely on tools and on which.
> > - The survey discusses the Toolformer (Schick et al. 2023) approach, which relies on the ALM to self-annotate useful tool uses in terms of how lower was the model’s perplexity with tool use vs. without. This work could be extended towards sequential tool use.
> > - We believe that taking the cost of using a tool into account can help to teach ALMs when to use a tool and which (see above).
> > - Other very recent approaches propose embedding for tools, allowing to know which to use among a great number of different and potentially overlapping APIs. Inspiration on how to combine the output of tools that overlap in capabilities can also be found in the data integration literature (Halevy and Dwivedi-Yu, 2023).
> >
> > *Combining reasoning and tool use.*
> >
> > It is not clear whether theorem provers lend themselves to non-algorithmic tasks, in particular, when a satisfactory answer cannot be unambiguously evaluated (think about text generation). A promising direction is to align reasoning steps with tool use, meaning that at most one tool should be required to validate a reasoning step. This could be done through fine-tuning from annotations, and can already be done via prompting: some LLMs such as GPT-4 exhibit such capabilities in a zero-shot fashion for some tools and some problems when given some brief explanation on the tool. However, there is little to no evaluation of such capabilities, especially for more complex tasks and tool combinations: instead of thinking about new methods, we believe the current most important thing to do is building a benchmark dedicated to solving problems that potentially require complex combinations of tools.
> >
> > *RL.*
> >
> > We added the following discussion to section 4.3:
> >
> > The promise of these methods is that a dataset can be generated via feedback and interactions of any behavior policy and can be used to train an improved policy. Therefore, combining a data-driven approach with offline RL could give the "best of both worlds": learning from counterfactual events in a scalable and stable way \citep[levine2020offline]. In the offline regime, though, there are some open challenges. The maximum improvement can be limited by a number of factors such as: the suboptimality of the initial behavior policy, the dimensionality of the state and the action space, the length of the effective horizon, the accumulation of errors and distributional shifts due to the discrepancy between the behavior policy and the learned one. \citep[levine2020offline]
> >
> > Levine et al. (2020): Offline Reinforcement Learning: Tutorial, Review, and Perspectives on Open Problems
> >
> > **On moving away from language modeling**
> >
> > Indeed, the choice of a tool depends on its context and we modified the equation to better reflect that. However, we do not fully agree with the reviewer’s claim that tool use is tractable, and give two counter arguments:
> >
> > - In the general case, marginalizing over all the tools is not enough, one also has to marginalize over all possible tool use.  For many tools however, the probability p(c | x1,..., x_t-1) associated with each possible tool output cannot be modeled. Take for example web browsing: for a fixed query, the result may vary day to day unpredictably as both the internet and the search algorithms are constantly evolving, making the evaluation of p(c | x1, … x_t-1) impossible unless the tool uses a past snapshot of the internet, which is generally not wanted.
> > - In the case where a tool can be called within another tool call, the number of tool uses can become exponential in the number of tools. For example, within a tool call, an ALM could browse the web by reading the content of a web page, which is text, then apply another of the N tools to the text it found to refine its query etc. If for some reason we require a depth that is equal to the number of tools, we have an exponential complexity.
> >
> > We added these points in the revised version of the survey.

---

> > > ### Comment · Reviewer_7WnD · 2023-06-13
> > > **Thank You for the Detailed Response**
> > >
> > > Thank you for the detailed response! I am glad to see that these comments have been addressed and integrated into the revised version.
> > >
> > > Also, great point on why tool use is intractable, despite a seemingly tractable surface form, due to potential tool use recursions, etc. It would be great to see this point outlined more explicitly in the paper to avoid similar confusions.
> > >
> > > Most of my comments, questions, and suggestions have therefore been resolved.

---

### Review · Reviewer_Nz7C · 2023-05-07

**Summary Of Contributions:**

This paper presents a comprehensive survey on augmented language models (ALMs). Maybe slightly different from others’ definition of ALMs, the authors define ALMs as LMs augmented with reasoning and tools, resulting in agents that combine reasoning, tool usage, and action, in addition to purely LMs. In this regard, the survey systematically reviews ALMs with respect to reasoning, tool usage, and action, which were usually discussed separately before. This approach offers a more visionary perspective for readers to view LLMs and gain a deeper understanding of the underlying connections among recent advancements in these fields.

**Audience:**

Yes

**Broader Impact Concerns:**

This paper included a nice discussion on ethical concerns that I feel sufficient.


**Claims And Evidence:**

Yes

**Requested Changes:**

I don't have requested changes for this paper.

**Strengths And Weaknesses:**

### Strengths:

1. This paper provides a novel perspective on augmented language models – by reviewing reasoning, tool usage, and action in the same framework, ALMs are envisioned as powerful agents. The paper also includes an interesting discussion on the connection between ALMs and world models. Although LLMs as agents are not new, this paper clearly defines and reviews the critical aspects of serving as agents, which I appreciate a lot by reading the paper.
2. This paper includes a comprehensive list of related work on reasoning, tool usage, and action, which should be helpful for researchers in related fields.


### Weaknesses:

I don’t identify significant weaknesses of this paper.

---

> ### Author Response · Authors · 2023-05-25
>
> We thank the reviewer for their positive feedback.

---

### Review · Reviewer_SRe8 · 2023-05-13

**Summary Of Contributions:**

This survey paper reviews papers where language models are augmented with using tools and reasoning. These models are dubbed ALMs and recent advances in this paradigm are reviewed in the paper. The paper is organized well into

- motivation, definitions,
- dedicated sections on reasoning and tools+actions,
- section on learning techniques to achieve these capabilities
- discussion and benefits sections

**Audience:**

Yes

**Broader Impact Concerns:**

As mentioned above, LLMs especially ones that are augmented to use tools and act on the results are becoming extremely popular in the current world. A few large companies have the capability to build the large pre-trained models but there is also a sprawling open-source ecosystem that’s using these LLMs to build products in the real world. This paper would benefit from brief commentary on that topic.

**Claims And Evidence:**

Yes

**Requested Changes:**

Section 2.1: I believe the scratch pad paper and Cobbe et al proposed the original ideas here. Please see this thread for relevant division: https://twitter.com/gstsdn/status/1533841505172922369?s=46&t=YDJSnSYS3msxPvOtE0K2Og
And fix the citations and language to reflect this.

Section 2.2: The explanation for Patel et al. 2022 is incomplete and hard to understand. Please rephrase.

Section 3.2.1: Explanation for RETRO can be made clearer with a few more sentences.

Section 3.4 typo: “a framework to gather an integrate” change an to “and”

Ethical concerns section could have been further enhanced to reflect the real-world impact these approaches are having in the developer community.

**Strengths And Weaknesses:**

Strengths:
- The paper is clearly written and is easy to read. The organization of the paper into dedicated sections for reasoning, tools and action are useful.
- The paper does a good job of comparing various works wherever possible like table 1 and 2.
- The figures with example prompts for important methods are clear to parse and understand and they add value to the narrative.
- The section of various learning techniques to achieve language model augmentation are useful.

Weaknesses:
- The ethical concerns section could have gone into more detail instead of a single paragraph.
- The paper could have benefited from adding more comparative results across different approaches. Tables 1 and 2 were informative in this regard but more of this could have helped.

---

> ### Author Response · Authors · 2023-05-25
>
> We thank the reviewer for their positive feedback. We provide a list of the changes made to the submission in light of the reviewer’s comments:
>
> - We give more credit to Cobbe et al. and Nye et al. in the step-by-step reasoning section.
> - We provide better explanations of Patel et al., and Borgeaud et al.
> - The ethical concern section has been strengthened (see general answer).
>
> Including tables of comparison requires a strong common ground between different methods, such as a benchmark (Table 1), or a general pipeline (Table 2): we did not see fit for tables in other parts of the survey since we gather different methods pursuing different goals, e.g in the reinforcement learning section. However, we would happily add a Table on a given theme if the reviewer thinks a particular section deserves more comparison.

---

### Author Response · Authors · 2023-05-25
**General answer**

We thank the reviewers for their positive comments. We provide a general answer below before addressing each reviewer’s comment individually.

**SRe8, 7WnD on strengthening the ethical section.**

We agree it is worth commenting the impact of ALMs in more depth, especially in light of the last months. We updated the ethical concerns section accordingly:

ALMs raise new ethical concerns. LM predictions based on tools may look more trustworthy and authoritative at a first glance, when in fact many of them will still be incorrect. Moreover, we can expect this phenomenon to be amplified as LMs reason in quite a similar manner to humans (Dasgupta et al., 2022), making it even harder to detect mistakes. Hence, ALMs may be leveraged to generate more convincing fake news and conspiracy theories. Conversely, conditioning ALMs on “safe” and trusted sources should lead to more factuality and less toxicity in their answers.

While such ethical concerns apply to most tools, it is important to distinguish between passive and active tools. The former only collects external information and passes it to the LM’s context, while the latter allows it to act on the virtual or physical world without human validation in the loop. An example of  active tool is letting the LM control a search engine. Hence, there exists a broad spectrum of possible harmful consequences of LM usage. We are moving from passive LMs that generate text in isolation of the external environment, towards ALMs that act in the real world. In this context, the aforementioned ethical concerns may resonate even further, as ALM will be connected to more and more tools and environments.

Currently, training LLM results in a large number of gas emissions, leveraging tools may result in smaller hence cheaper models that are more environmentally friendly both at training, inference, and update. ALMs are therefore a promising avenue to decrease the environmental impact of LLMs.


ALMs also have the potential to transform the LM landscape. Nowadays, state of the art LLMs have been the product of large research organizations, fine-tuning these models towards more versatility (via reasoning and learning to use different tools) and agency (via actions) may be within range of smaller entities and perhaps individuals. We may therefore see the burgeoning of a market of open, possibly specialized and even personalized ALM agents, learning to use new tools, handling various data modalities and interacting with each other. We may also see the emergence of a new breed of app store, dedicated (via appropriate documentation or API constraints for example) to be used by ALMs. Overall, this new ecosystem may further accelerate the current pace of LLM research and application to the real world, while also making its control even more uncertain.

---

### Decision · Action_Editors · 2023-06-24

**Recommendation:** Accept as is

**Comment:**

The paper provides a good review of a very fast-moving field, with a lot of good references.  I enjoyed reading it and all reviewers recommended acceptance.  It is also very appropriate for a survey certification.

There is a small issue in Figure 6, where the answer line seems incorrect, and should be `answer = tennis_balls + bought_balls` instead of `tennis_balls * bought_balls`, I don’t know if this is a typo or intentional.

**Audience:**

A lot of researchers and practitioners working on language modeling and general AI assistants would benefit from such a survey of this fast-moving field.

**Claims And Evidence:**

This paper presents a survey of augmented language models (ALMs) that reasons, uses tools and acts in the virtual / physical world.  The contribution is a comprehensive survey of recent literature on this topic, and they have executed it quite well, reviewing the literature along a few dimensions, including (1) capabilities, i.e. what kind of new capability can the ALMs have, on reasoning / tool use / action, and (2) how to implement the new capabilities, e.g. through few-shot prompting, fine-tuning or RL.  The paper also presented a few questions for discussion and directions for future research.